# RETHINKING UNCERTAINTY ESTIMATION IN NATURAL LANGUAGE GENERATION

## ABSTRACT

Large language models (LLMs) are increasingly employed in real-world applications, driving a need to determine when their generated text can be trusted or should be questioned. To assess the trustworthiness of the generated text, reliable uncertainty estimation is essential. Current LLMs generate text through a stochastic process that can lead to different output sequences for the same prompt. Consequently, leading uncertainty measures require generating multiple output sequences to estimate the LLM's uncertainty. However, generating additional output sequences is computationally expensive, making these uncertainty estimates impractical at scale. In this work, we challenge the theoretical foundations of the leading measures and derive an alternative measure that eliminates the need for generating multiple output sequences. Our new measure is based solely on the negative log-likelihood of the most likely output sequence. This vastly simplifies uncertainty estimation while maintaining theoretical rigor. Empirical results demonstrate that our new measure achieves state-of-the-art performance across various models and tasks. Our work lays the foundation for reliable and efficient uncertainty estimation in LLMs, challenging the necessity of the more complicated methods currently leading the field.

## 1 INTRODUCTION

Language models are increasingly adopted in a wide range of real-world applications. Despite the advancements in language models, determining whether a generated text can be trusted remains a significant challenge. To address this challenge, it is crucial to reliably assess the level of certainty a language model has regarding its generated text. While uncertainty estimates do not guarantee factuality for generated text based on consistent but erroneous training data, they are a reliable indicator of errors at present (Kuhn et al., 2023; Aichberger et al., 2024; Farquhar et al., 2024).

Assessing predictive uncertainty in language models is inherently difficult due to their autoregressive nature. For a given input sequence, language models predict the next token probabilities, based on which a specific token is selected and appended to the sequence. This stochastic process is repeated for each new token. Selecting different tokens at specific steps during the generation leads to varying output sequences for the same input sequence with the same language model. Consequently, the space of possible output sequences is vast and computationally intractable to fully explore (Sutskever et al., 2014; Vaswani et al., 2017; Radford et al., 2018).

Current uncertainty estimation methods rely on assessing the probability distribution over all possible output sequences. However, the generation of each additional token is computationally expensive, and practical methods can only sample a small fraction of possible output sequences (Malinin & Gales, 2021; Kadavath et al., 2022). Moreover, even after having generated multiple likely output sequences, the question remains whether these indicate high uncertainty. A language model that likely generates different output sequences is not necessarily uncertain about the underlying meaning if the output sequences are semantically equivalent. Leading uncertainty measures address this fact by considering the semantics of the output sequences, utilizing separate language inference models (Kuhn et al., 2023; Farquhar et al., 2024). While these measures improve the performance of the uncertainty estimates, they also further add complexity and computational overhead. These factors make current uncertainty estimation methods impractical at scale, hindering their broad adoption in

real-world applications. There is a need for efficient uncertainty estimation methods that give clear insights into the reliability of language models without incurring substantial computational costs.

In this work, we assess whether we can theoretically motivate an uncertainty measure that does not rely on the probability distribution over all possible output sequences. Building on insights from the principled framework of proper scoring rules (Kotelevskii & Panov, 2024; Hofman et al., 2024), we adopt the zero-one score as an alternative to the currently used logarithmic score for uncertainty measures in NLG. This leads to a theoretically motivated measure that does not require generating multiple output sequences but solely relies upon a single output sequence. Our proposed measure is straightforward: it simply is the negative log-likelihood of the most likely output sequence. By eliminating the need to generate and semantically cluster multiple output sequences, our measure significantly reduces computational costs and complexity.

Experimental results demonstrate that our new measure matches or even exceeds the performance of current state-of-the-art uncertainty estimation methods across various model classes, model sizes, model stages, tasks, datasets, and evaluation metrics. In summary, our new measure not only preserves theoretical rigor but also provides a more scalable solution for uncertainty estimation in language models, making it highly practical for real-world applications.

Our main contributions are:
- We introduce the negative log-likelihood of the most likely output sequence as an efficient and practical measure of uncertainty in NLG.
- We provide a rigorous theoretical foundation for our measure, building upon established principles in uncertainty theory and proper scoring rules.
- We conduct extensive experiments demonstrating that our measure achieves strong performance, matching or surpassing state-of-the-art methods while significantly reducing computational costs.

## 2 PREDICTIVE UNCERTAINTY IN NLG

**Preliminaries.** We assume a fixed training dataset $\mathcal{D} = \{s_i\}_{i=1}^N$ consisting of ordered tokens $s_t \in \mathcal{V}$, with $\mathcal{V}$ being a given vocabulary. Each token at step $t$ is assumed to be sampled according to the predictive distribution $p(s_t \mid s_{<t}, w^*)$, conditioned on the sequence of preceding tokens $s_{<t}$ and the true (but unknown) language model parameters $w^*$. We assume that the given model class can theoretically represent the true predictive distribution, a common and usually necessary assumption (Hüllermeier & Waegeman, 2021). How likely language model parameters $\tilde{w}$ match $w^*$ is determined by the posterior distribution $p(\tilde{w} \mid \mathcal{D}) = p(\mathcal{D} \mid \tilde{w})p(\tilde{w})/p(\mathcal{D})$.

In language model inference, the input to a given language model parameterized by $w$ is a sequence $x = (x_1, ..., x_M)$ and the output is a sequence $y = (y_1, ..., y_T) \in \mathcal{Y}_T$, with $x, y \in \mathcal{V}$ and $\mathcal{Y}_T$ being the set of all possible output sequences with a sequence length smaller equal to $T$. The likelihood of a token $y_t \in y$ being generated by the language model is conditioned on both the input sequence and all previously generated tokens, denoted as $p(y_t \mid x, y_{<t}, w)$. The likelihood of output sequences $y \in \mathcal{Y}_T$ being generated by the language model is then the product of the individual token probabilities, denoted as $p(y \mid x, w) = \prod_{t=1}^T p(y_t \mid x, y_{<t}, w)$ (Sutskever et al., 2014), while the heuristic length-normalized variant is $\bar{p}(y \mid x, w) = \exp\left(\frac{1}{T}\sum_{t=1}^T \log p(y_t \mid x, y_{<t}, w)\right)$ (Malinin & Gales, 2021).

Computing the likelihood of a specific output sequence $y$ being generated by the language model parameterized by $w$ – or in other words, being sampled from the probability distribution over possible output sequences $y \sim p(y \mid x, w)$ – is straightforward. The language model directly provides the individual token likelihoods. However, determining the full probability distribution over possible output sequences is considerably more challenging, since $\mathcal{Y}_T$ scales exponentially with the sequence length $T$. The computational complexity of evaluating all possible sequences grows as $\mathcal{O}(|\mathcal{V}|^T)$. Since modern language models even exceed a vocabulary size $|\mathcal{V}|$ of one hundred thousand tokens, this distribution becomes intractable to compute, even for relatively short maximal sequence lengths $T$ (Dubey et al., 2024).

**Uncertainty Measures and Proper Scoring Rules.** We now derive measures to estimate uncertainty in NLG. Throughout this work, the focus is on estimating the predictive uncertainty of a single, given "off-the-shelf" model. We assume to that a given language model parameterized by $\boldsymbol{w}$ is the *predicting model* used to sample output sequences $\boldsymbol{y} \sim p(\boldsymbol{y} \mid \boldsymbol{x}, \boldsymbol{w})$. Furthermore, we assume that any language model parameterized by $\tilde{\boldsymbol{w}}$ is an *approximation of the true predictive distribution* according to its posterior probability $p(\tilde{\boldsymbol{w}} \mid \mathcal{D})$. Together, these two assumptions give rise to specific uncertainty measures (Schweighofer et al., 2023; 2024), as elaborated on in more detail below. Aichberger et al. (2024) shows that established uncertainty measures in NLG, such as Predictive Entropy (PE) (Malinin & Gales, 2021) and Semantic Entropy (SE) (Kuhn et al., 2023; Farquhar et al., 2024), naturally emerge under this assumption. In general, the information-theoretic entropy has become the standard measure to assess predictive uncertainty. However, recent studies by Lahlou et al. (2023); Gruber & Buettner (2023); Kotelevskii & Panov (2024) and Hofman et al. (2024) have shown that these information-theoretic measures are not the only viable options. A broader class of *proper scoring rules* provides a principled framework for predictive uncertainty measures. In the following, we leverage this framework to derive our alternative measure that relies solely on a single output sequence. We begin by discussing the concept of proper scoring rules.

In general, proper scoring rules are a class of functions that evaluate the quality of probabilistic predictions by assigning a numerical score based on the predictive distribution and the actual observations (Gneiting & Raftery, 2007). For uncertainty estimation in NLG, the general notion of proper scoring rules assigns a numerical score to how well the predicted distribution of output sequences $p(\boldsymbol{y} \mid \boldsymbol{x}, \cdot)$ aligns with the observed output sequence $\boldsymbol{y}'$. In particular, a proper scoring rule is an extended real-valued function $\mathrm{S} : \mathcal{P} \times \mathcal{Y} \to [-\infty, \infty]$, such that $\mathrm{S}(p, \cdot)$ is $\mathcal{P}$-quasi-integrable over a convex class of probability measures $\mathcal{P}$. The expected score over possible output sequences $\boldsymbol{y}'$ is given by

$$\mathrm{E}_{\boldsymbol{y}' \sim p(\boldsymbol{y}' \mid \boldsymbol{x}, \cdot)} \left[ \mathrm{S} \left( p(\boldsymbol{y} \mid \boldsymbol{x}, \cdot), \boldsymbol{y}' \right) \right] \tag{1}$$

Given this general formulation, we now incorporate the assumptions outlined above to establish the connection to uncertainty measures (Schweighofer et al., 2024). First, under the assumption about the *predicting model*, the distribution giving rise to the observed output sequences $p(\boldsymbol{y}' \mid \boldsymbol{x}, \cdot)$ corresponds to the predictive distribution of the given language model, denoted as $p(\boldsymbol{y}' \mid \boldsymbol{x}, \boldsymbol{w})$. Second, under the assumption about the *approximation of the true predictive distribution*, a sampled output sequence $\boldsymbol{y}'$ has to be compared to all possible language models parameterized by $\tilde{\boldsymbol{w}}$, according to their posterior distribution $p(\tilde{\boldsymbol{w}} \mid \mathcal{D})$. This captures how much the sampled output sequence aligns with all possible predictive distributions $p(\boldsymbol{y} \mid \boldsymbol{x}, \tilde{\boldsymbol{w}})$. Therefore, we take a posterior expectation over Eq. (1), which can be additively decomposed into an entropy term and a divergence term (Gneiting & Raftery, 2007; Kull & Flach, 2015):

$$\underbrace{\mathrm{E}_{\tilde{\boldsymbol{w}} \sim p(\tilde{\boldsymbol{w}} \mid \mathcal{D})} \left[ \mathrm{E}_{\boldsymbol{y}' \sim p(\boldsymbol{y}' \mid \boldsymbol{x}, \boldsymbol{w})} \left[ \mathrm{S} \left( p(\boldsymbol{y} \mid \boldsymbol{x}, \tilde{\boldsymbol{w}}), \boldsymbol{y}' \right) \right] \right]}_{\text{expected score}} \tag{2}$$

$$= \underbrace{\mathrm{E}_{\boldsymbol{y}' \sim p(\boldsymbol{y}' \mid \boldsymbol{x}, \boldsymbol{w})} \left[ \mathrm{S} \left( p(\boldsymbol{y} \mid \boldsymbol{x}, \boldsymbol{w}), \boldsymbol{y}' \right) \right]}_{\text{entropy term}}$$

$$+ \underbrace{\mathrm{E}_{\tilde{\boldsymbol{w}} \sim p(\tilde{\boldsymbol{w}} \mid \mathcal{D})} \left[ \mathrm{E}_{\boldsymbol{y}' \sim p(\boldsymbol{y}' \mid \boldsymbol{x}, \boldsymbol{w})} \left[ \mathrm{S} \left( p(\boldsymbol{y} \mid \boldsymbol{x}, \tilde{\boldsymbol{w}}), \boldsymbol{y}' \right) - \mathrm{S} \left( p(\boldsymbol{y} \mid \boldsymbol{x}, \boldsymbol{w}), \boldsymbol{y}' \right) \right] \right]}_{\text{divergence term}} .$$

**Aleatoric and Epistemic Uncertainty.** In terms of predictive uncertainty, this general framework can be interpreted as follows. The expected score over possible output sequences and language model parameters captures the *total* uncertainty of the given language model. The entropy term reflects *aleatoric* uncertainty, which is the uncertainty inherent in the data generation process, arising from the inherent variability and randomness in natural language (Gal, 2016; Kendall & Gal, 2017). The divergence term reflects *epistemic* uncertainty, which quantifies the uncertainty due to lack of knowledge about the true language model parameters, arising from limited data or model capacity (Houlsby et al., 2011; Gal, 2016; Malinin, 2019; Hüllermeier & Waegeman, 2021).

The concrete *total*, *aleatoric*, and *epistemic* uncertainty measures depends on the choice of proper scoring rule. For instance, the logarithmic score is the most common proper scoring rule that gives rise to the well-known information-theoretic uncertainty measures in both classification tasks (Houlsby et al., 2011; Gal, 2016) and NLG (Malinin & Gales, 2021; Kuhn et al., 2023).

In the following, we first revisit these uncertainty measures that are based on the logarithmic score and analyze their effectiveness in estimating aleatoric and epistemic uncertainty. Thereafter, we propose uncertainty measures that are based on another proper scoring rule, the zero-one score. This score has not yet been considered for uncertainty estimation in NLG. We show that utilizing uncertainty measures based on the zero-one score offers certain advantages.

## 2.1 Established Uncertainty Measures in NLG based on Logarithmic Score

The logarithmic score is usually assumed implicitly to derive uncertainty measures, due to the foundation of resulting measures in information theory (Lahlou et al., 2023; Gruber & Buettner, 2023; Hofman et al., 2024; Kotelevskii & Panov, 2024). In the context of NLG, it considers the negative log-likelihood of a generated output sequence $\boldsymbol{y}'$:

$$\mathrm{S}_{\log}\left(p(\boldsymbol{y} \mid \boldsymbol{x}, \cdot), \boldsymbol{y}'\right) = -\log p(\boldsymbol{y} = \boldsymbol{y}' \mid \boldsymbol{x}, \cdot) . \tag{3}$$

Using the logarithmic score in Eq. (2) results in the cross-entropy $\mathrm{CE}(\cdot\,;\cdot)$ between the output sequence distribution of the given language model and that of every possible language model according to their posterior $p(\tilde{\boldsymbol{w}} \mid \mathcal{D})$ (Schweighofer et al., 2023; Aichberger et al., 2024):

$$\underbrace{\mathrm{E}_{\tilde{\boldsymbol{w}}\sim p(\tilde{\boldsymbol{w}}|\mathcal{D})}\left[\mathrm{CE}(p(\boldsymbol{y} \mid \boldsymbol{x}, \boldsymbol{w}); p(\boldsymbol{y} \mid \boldsymbol{x}, \tilde{\boldsymbol{w}}))\right]}_{\text{total}} \tag{4}$$

$$= \underbrace{\mathrm{H}(p(\boldsymbol{y} \mid \boldsymbol{x}, \boldsymbol{w}))}_{\text{aleatoric}} + \underbrace{\mathrm{E}_{\tilde{\boldsymbol{w}}\sim p(\tilde{\boldsymbol{w}}|\mathcal{D})}\left[\mathrm{KL}(p(\boldsymbol{y} \mid \boldsymbol{x}, \boldsymbol{w}) \,\|\, p(\boldsymbol{y} \mid \boldsymbol{x}, \tilde{\boldsymbol{w}}))\right]}_{\text{epistemic}} .$$

The epistemic uncertainty is a posterior expectation of the Kullback-Leibler divergence $\mathrm{KL}(\cdot \,\|\, \cdot)$ between the output sequence distribution of the given model and that of all possible models. This requires considering every possible model parametrization. Since modern language models have billions of parameters (Radford et al., 2018; Zhang et al., 2022; Touvron et al., 2023; Zuo et al., 2024; Dubey et al., 2024), the epistemic uncertainty is particularly challenging to estimate.

Current work usually solely considers the aleatoric uncertainty, which is the Shannon entropy $\mathrm{H}(\cdot)$ of the output sequence distribution of the given language model (Malinin & Gales, 2021; Kuhn et al., 2023; Aichberger et al., 2024). Computing the output sequence distribution still requires considering the whole set of possible output sequences $\mathcal{Y}_T$. Thus, the primary objective of uncertainty estimation based on the logarithmic score is to closely approximate this output sequence distribution.

**Predictive Entropy.** The aleatoric uncertainty under a given language model is the entropy of the output sequence distribution, commonly referred to as Predictive Entropy (PE). Intuitively, high PE implies that the language model is likely to generate different output sequences from the same input sequence, indicating high uncertainty of the language model. PE usually is estimated via Monte Carlo (MC) sampling (Malinin & Gales, 2021):

$$\mathrm{H}(p(\boldsymbol{y} \mid \boldsymbol{x}, \boldsymbol{w})) = \mathrm{E}_{\boldsymbol{y}\sim p(\boldsymbol{y}|\boldsymbol{x},\boldsymbol{w})}\left[-\log p(\boldsymbol{y} \mid \boldsymbol{x}, \boldsymbol{w})\right] \tag{5}$$

$$\approx \frac{1}{N}\sum_{n=1}^{N} -\log p(\boldsymbol{y}^n \mid \boldsymbol{x}, \boldsymbol{w}) , \qquad \boldsymbol{y}^n \sim p(\boldsymbol{y} \mid \boldsymbol{x}, \boldsymbol{w}) .$$

**Semantic Entropy.** Semantic Entropy (SE) builds on the fact that output sequences may be different on a token level but equivalent on a semantics level. In such cases, the PE can be misleading, as it reflects high uncertainty even when different output sequences have the same semantic meaning. PE also captures the uncertainty of the language model in expressing the semantically same statement, which is often not the focus of uncertainty estimation in NLG. Thus, instead of the entropy of the output sequence distribution, the entropy of the semantic cluster distribution is considered, denoted as $p(c \mid \boldsymbol{x}, \boldsymbol{w}) = \sum_{\mathcal{Y}} p(c \mid \boldsymbol{x}, \boldsymbol{y}, \boldsymbol{w})\, p(\boldsymbol{y} \mid \boldsymbol{x}, \boldsymbol{w})$. The probability of an output sequence belonging to a semantic cluster is usually approximated with a separate natural language inference model. High SE implies that the language model is likely to generate output sequences that have high semantic diversity, indicating high semantic uncertainty (Kuhn et al., 2023; Farquhar et al., 2024).

$$\mathrm{H}(p(c \mid \boldsymbol{x}, \boldsymbol{w})) = \mathrm{E}_{c\sim p(c|\boldsymbol{x},\boldsymbol{w})}\left[-\log p(c \mid \boldsymbol{x}, \boldsymbol{w})\right] \tag{6}$$

$$\approx \frac{1}{N}\sum_{n=1}^{N} -\log p(c^n \mid \boldsymbol{x}, \boldsymbol{w}) , \qquad c^n \sim p(c \mid \boldsymbol{x}, \boldsymbol{w}) .$$

Each of these uncertainty measures based on the logarithmic score considers the distribution over all possible output sequences $p(\boldsymbol{y} \mid \boldsymbol{x}, \boldsymbol{w})$, which is defined over the entire set of possible output sequences $\mathcal{Y}_T$. To approximate this distribution, it requires sampling output sequences from $\mathcal{Y}_T$. This requires generating multiple output sequences, which is computationally expensive. In the following, we eliminate this requirement by considering an alternative proper scoring rule.

## 2.2 NEW UNCERTAINTY MEASURES IN NLG BASED ON ZERO-ONE SCORE

Next, we introduce measures based on the zero-one score, which has not yet been considered as a proper scoring rule for deriving uncertainty measures in NLG. The zero-one score considers the predictive distribution for the most likely output sequence:

$$
\mathrm{S}_{\text{0-1}}\left(p(\boldsymbol{y} \mid \boldsymbol{x}, \cdot), \boldsymbol{y}'\right) = \begin{cases} 1 - p(\boldsymbol{y} = \boldsymbol{y}' \mid \boldsymbol{x}, \cdot) & \text{if } \boldsymbol{y}' = \operatorname{argmax}_{\boldsymbol{y}} p(\boldsymbol{y} \mid \boldsymbol{x}, \cdot), \\ 0 & \text{otherwise.} \end{cases} \tag{7}
$$

Using the zero-one score in Eq. (2) results in the total uncertainty being the expected confidence of the given language model about the most likely output sequences generated by all language models according to their posterior probability $p(\boldsymbol{w} \mid \mathcal{D})$:

$$
\underbrace{\mathrm{E}_{\tilde{\boldsymbol{w}} \sim p(\tilde{\boldsymbol{w}} \mid \mathcal{D})}\left[1 - p(\boldsymbol{y} = \tilde{\boldsymbol{y}}^* \mid \boldsymbol{x}, \boldsymbol{w})\right]}_{\text{total}} \tag{8}
$$

$$
= \underbrace{1 - p(\boldsymbol{y} = \boldsymbol{y}^* \mid \boldsymbol{x}, \boldsymbol{w})}_{\text{aleatoric}} + \underbrace{p(\boldsymbol{y} = \boldsymbol{y}^* \mid \boldsymbol{x}, \boldsymbol{w}) - \mathrm{E}_{\tilde{\boldsymbol{w}} \sim p(\tilde{\boldsymbol{w}} \mid \mathcal{D})}\left[p(\boldsymbol{y} = \tilde{\boldsymbol{y}}^* \mid \boldsymbol{x}, \boldsymbol{w})\right]}_{\text{epistemic}},
$$

with $\boldsymbol{y}^* = \operatorname{argmax}_{\boldsymbol{y}} p(\boldsymbol{y} \mid \boldsymbol{x}, \boldsymbol{w})$ and $\tilde{\boldsymbol{y}}^* = \operatorname{argmax}_{\boldsymbol{y}} p(\boldsymbol{y} \mid \boldsymbol{x}, \tilde{\boldsymbol{w}})$. Similar to Eq. (4), the epistemic uncertainty is a posterior expectation that remains challenging to estimate. However, we again focus on the aleatoric uncertainty, which solely considers the likelihood of the most likely output sequence under the given language model.

While aleatoric uncertainty derived from the logarithmic score requires approximating the entire output sequence distribution by sampling multiple sequences (as seen in Eq. (5) and Eq. (6)), the aleatoric uncertainty based on the zero-one score (see Eq. (8)) requires approximating the most likely output sequence under the given language model. This distinction is crucial, as approximating the most likely output sequence aligns directly with standard inference techniques widely used in language models, such as greedy decoding, beam search (Sutskever et al., 2014), top-k sampling, or nucleus sampling (Holtzman et al., 2020). For numerical stability, we consider the negative log-likelihood of the most likely output sequence that is proportional to the measure of aleatoric uncertainty in Eq. (8). We propose to estimate this quantity using the greedily decoded output sequence as an efficient and effective measure of aleatoric uncertainty:

$$
\mathrm{NLL} := -\sum_{t=1}^{T} \log\left(\max_{y_t} p(y_t \mid \boldsymbol{x}, \boldsymbol{y}_{<t}, \boldsymbol{w})\right) \approx -\log p(\boldsymbol{y} = \boldsymbol{y}^* \mid \boldsymbol{x}, \boldsymbol{w}) \tag{9}
$$

**Discussion.** Our proposed uncertainty measure challenges the prevailing reliance on multi-sequence sampling and semantic clustering for uncertainty estimation in NLG. By solely relying on the output sequences generated with greedy decoding, our approach significantly reduces computational overhead while maintaining theoretical rigor through its foundation in proper scoring rules. While uncertainty measures based on the logarithmic score could theoretically excel if the full distribution over output sequences $p(\boldsymbol{y} \mid \boldsymbol{x}, \boldsymbol{w})$ were accessible – as in standard classification tasks – this distribution is intractable for NLG tasks due to their sequential nature. As a result, sampling-based methods often yield crude approximations, constrained by computational limits and sampling variability. In contrast, our uncertainty measure, based on the zero-one score, offers a more rigorous alternative while eliminating the need for extensive sampling. In Sec. 4, we demonstrate that using our measure of uncertainty yields performance that is superior to or at least on par with uncertainty measures based on the logarithmic score. This makes our method more practical for large-scale applications.

## 3 RELATED WORK

In the previous section, we discussed uncertainty estimation methods based on the logarithmic score. Beyond these, there is a body of work that extends the concept of Semantic Entropy (Kuhn et al., 2023; Farquhar et al., 2024), for instance by either improving the semantic clustering (Nikitin et al., 2024; Qiu & Miikkulainen, 2024), improving the sampling of output sequences (Aichberger et al., 2024), or directly approximating the measure from hidden states of the language model (Kossen et al., 2024). Also, there is a body of work that builds upon the concept of Predictive Entropy (Malinin & Gales, 2021), for instance by considering a weighting factor for individual token and sequence likelihoods to account for the importance on a semantic level (Duan et al., 2023; Bakman et al., 2024).

There is also work on uncertainty estimation in NLG that is not grounded in proper scoring rules. For instance, several approaches leverage the language model itself to directly predict uncertainty, whether through numerical estimates or verbal explanations (Mielke et al., 2022; Lin et al., 2022; Kadavath et al., 2022; Cohen et al., 2023a; Ganguli et al., 2023; Ren et al., 2023; Tian et al., 2023). Cohen et al. (2023b) employ cross-examination, where one language model generates an output sequence and another model acts as an examiner to assess uncertainty. Zhou et al. (2023) explore the behavior of language models when expressing their uncertainty, providing insights into how models articulate confidence in their predictions. Also, Manakul et al. (2023) propose using sampled output sequences as input for another language model to assess uncertainty, offering a unique perspective on sequence evaluation. Additionally, Xiao et al. (2022) provide an empirical analysis of how factors such as model architecture and training data influence uncertainty estimates. Finally, conformal prediction (Quach et al., 2023) offers another approach by calibrating a stopping rule for output sequence generation, providing a statistical framework for uncertainty estimation.

## 4 EXPERIMENTS

We aligned the evaluation of uncertainty estimation methods with related work by focusing on free-form question-answering tasks (Kuhn et al., 2023; Duan et al., 2023; Bakman et al., 2024; Nikitin et al., 2024; Aichberger et al., 2024; Kossen et al., 2024). While Farquhar et al. (2024) additionally concerns experiments with paragraph-length generations, their approach involves breaking down the entire paragraph into factual claims and reconstructing corresponding questions. Since the performance is expected to correlate with the performance on free-form question answering, we decided to focus specifically on free-form question answering tasks for a more direct assessment and less ambiguity in the evaluation.

**Datasets.** We evaluated uncertainty estimation methods on three different datasets. We used the over 3,000 test instances from *TriviaQA* (Joshi et al., 2017) concerning trivia questions, the over 300 test instances from *SVAMP* (Patel et al., 2021) concerning elementary-level math problems, and the over 3,600 test instances from *NQ-Open* (Lee et al., 2019) to assess natural questions aggregated from Google Search. Each dataset was utilized for two distinct tasks: (1) generating concise answers in the form of short phrases, and (2) producing more detailed answers in the form of full sentences (Farquhar et al., 2024). The resulting six tasks span a broad range of scenarios, ensuring a comprehensive evaluation of the uncertainty estimation methods.

**Models.** We conducted our evaluations on six distinct language models across different architectures, sizes, and training stages. Specifically, we used the Transformer-based model series *Llama-3.1* (Vaswani et al., 2017; Dubey et al., 2024) and the state-space model series *Falcon Mamba* (Gu & Dao, 2024; Zuo et al., 2024), representing two prominent language model paradigms. To assess the effect of training stage model scale on uncertainty estimation in NLG, we considered pre-trained (*PT*) and instruction-tuned (*IT*) language models with 7, 8, and 70 billion parameters, together covering a wide spectrum of model characteristics.

**Baselines.** We compare our method against the commonly used uncertainty measures based on the logarithmic score as of Eq. (5) and Eq. (6). These include Predictive Entropy (*PE*), length-normalized Predictive Entropy (*LN-PE*) (Malinin & Gales, 2021), Semantic Entropy (*SE*), length-normalized Semantic Entropy (*LN-SE*), and Discrete Semantic Entropy (*D-SE*) (Kuhn et al., 2023; Farquhar et al., 2024). For a given output sequence $\boldsymbol{y}'$, the length-normalized variants consider $\bar{p}(\boldsymbol{y}' \mid \boldsymbol{x}, \boldsymbol{w})$ instead of $p(\boldsymbol{y}' \mid \boldsymbol{x}, \boldsymbol{w})$ to compute the uncertainty estimates. The discrete variant of Semantic

Table 1: Average AUROC across TriviaQA, SVAMP and NQ datasets, using uncertainty estimates of different measures to distinguish between correct and incorrect answers. Varying model architectures (*transformer*, *state-space*), model sizes (*7B*, *8B*, *70B*), and model stages (*PT*, *IT*) are considered for generating answers. The reference answer is generated using *greedy decoding*, either as a whole sentence (*long*) or a short phrase (*short*). The reference answer's correctness is assessed by checking if the F1 score of the commonly used SQuAD metric exceeds 0.5 (*F1*) or if the LLM-as-a-judge considers it as correct (*LLM*). Predictive Entropy (*PE*), length-normalized Predictive Entropy (*LN-PE*), Semantic Entropy (*SE*), length-normalized Semantic Entropy (*LN-SE*), and discrete Semantic Entropy (*D-SE*) use 10 output sequences to assign an uncertainty estimate, each generated via multinomial sampling. NLL solely uses the reference answer to assign an uncertainty estimate.

| *Uncertainty measure based score* | | | | *Logarithmic* | | | | | *Zero-One* |
|---|---|---|---|---|---|---|---|---|---|
| **Model** | | **Gen.** | **Metric** | **PE** | **LN-PE** | **SE** | **LN-SE** | **D-SE** | **NLL** |
| Transformer | 8B | | short | F1 | .776 | .795 | .775 | .793 | .804 | **.824** |
| | | PT | short | LLM | .698 | .714 | .690 | .706 | .719 | **.726** |
| | | | long | LLM | .562 | .555 | .545 | .553 | .600 | **.649** |
| | | | short | F1 | .772 | .801 | .805 | .814 | .806 | **.838** |
| | | IT | short | LLM | .676 | .697 | .704 | .709 | .694 | **.722** |
| | | | long | LLM | .551 | .548 | .599 | .601 | .609 | **.615** |
| | 70B | | short | F1 | .775 | .790 | .793 | .803 | .791 | **.820** |
| | | PT | short | LLM | .693 | .709 | .718 | .722 | .715 | **.723** |
| | | | long | LLM | .552 | .534 | .558 | .569 | .571 | **.649** |
| | | | short | F1 | .748 | .781 | .790 | **.799** | .783 | .792 |
| | | IT | short | LLM | .681 | .698 | .703 | **.709** | .699 | .699 |
| | | | long | LLM | .555 | .557 | .568 | .595 | **.600** | .562 |
| State-Space | 7B | | short | F1 | .811 | .815 | .809 | .822 | .828 | **.843** |
| | | PT | short | LLM | .705 | .711 | .701 | .711 | .716 | **.728** |
| | | | long | LLM | .567 | .597 | .574 | .611 | **.624** | .612 |
| | | | short | F1 | .793 | .814 | .797 | .816 | .829 | **.838** |
| | | IT | short | LLM | .690 | .701 | .689 | .699 | .711 | **.719** |
| | | | long | LLM | .588 | .587 | .597 | .618 | **.629** | .615 |

Entropy entirely disregards the output sequence likelihood and only considers the proportion of output sequences that belong to the same semantic cluster (Farquhar et al., 2024).

**Evaluation.** Effective uncertainty measures should accurately reflect the reliability of answers generated by the language model. Higher uncertainty more likely leads to incorrect generations. Thus, to evaluate the performance of an uncertainty estimator, we assess how well it correlates with the correctness of the language model's answers; correct answers should be assigned a lower uncertainty estimator than incorrect answers. To determine whether an answer is correct, it has to be compared to the respective ground truth answer. To do so, we check if the F1 score of the commonly used SQuAD metric exceeds 0.5 (Rajpurkar et al., 2016). Although there are some limitations to using such a simple metric, it has relatively small errors in standard data sets and, therefore, remains widely used in practice However, this metric is only applicable for short-phrase generations that align with the ground truth answer. Therefore, we additionally employ Llama-3.1 with 70 billion parameters (Dubey et al., 2024) as an LLM-as-a-judge to assess the correctness of both short-phrase and full-sentence generations. Subsequently, to measure the correlation between incorrectness of answers and the respective uncertainty estimates, we use the Area Under the Receiver Operating Characteristic (AUROC). Higher AUROC values indicate better performance of the uncertainty estimator, as it reflects a stronger alignment between the correctness of the language model's answers and their respective uncertainty estimates. Overall, this evaluation process follows established methodologies for assessing the performance of uncertainty measures in NLG (Kuhn et al., 2023; Duan et al., 2023; Bakman et al., 2024; Farquhar et al., 2024; Nikitin et al., 2024; Aichberger et al., 2024; Kossen et al., 2024).

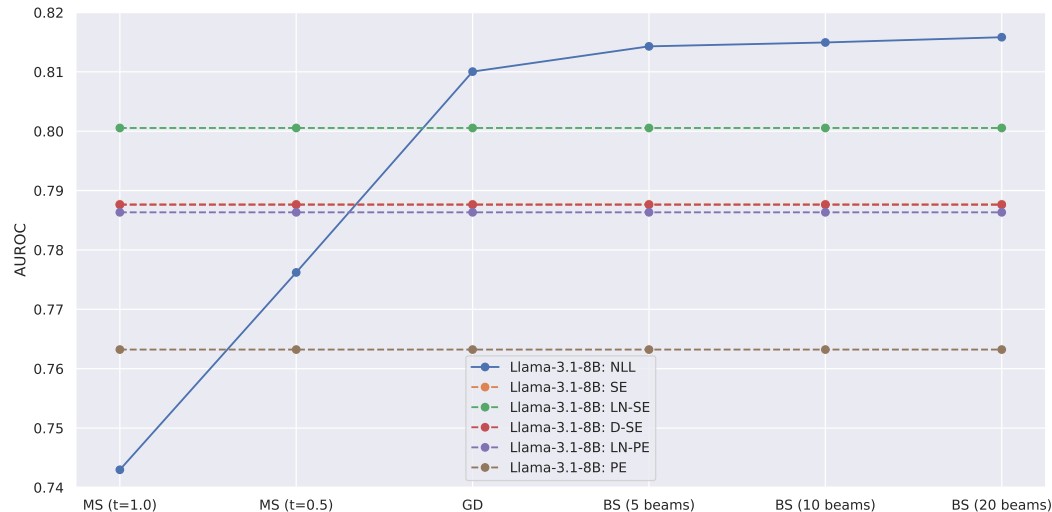

Figure 1: Average AUROC for the TriviaQA dataset, using the Llama-3.1-8B model to generate short phrase answers. The reference answer is generated using multinomial sampling (MS) with different temperature values (t), Greedy decoding (GD), and beam search (BS) with a different number of beams.

**Analysis of results.** Tab. 1 summarizes the performance of uncertainty measures across six different language models and six different tasks. Our proposed measure (NLL) largely outperforms current state-of-the-art uncertainty measures, particularly in tasks that involve generating short phrases. This suggests that our measure is highly effective when focusing on the critical part of the output sequence that contains the actual answer to a question. In practical scenarios, the reliability of the specific answer is often more relevant than the uncertainty of the entire generated sentence. Thus, our measure provides targeted and computationally efficient uncertainty estimates, delivering enhanced performance where it is most critical, especially in real-world applications.

**Approximating the most likely output sequence.** Figure 1 illustrates the performance of our uncertainty measure when considering different inference techniques for generating answers. The reference answer, generated via beam search with a size of 20, is used to assess correctness, as it provides the best approximation of the most likely answer generated by the language model. Since the baselines are evaluated on output sequences generated using their optimal hyperparameter settings, their performance remains consistent. The results show that as the approximation to the most likely answer improves, so does the performance of our measure. However, while multinomial sampling significantly degrades the performance of our uncertainty measure, greedy decoding achieves performance comparable to more precise methods, such as beam search, reinforcing its validity as an effective approximation of the most likely output sequence.

Further experimental results and insights into the behavior of the uncertainty estimators can be found in Sec. A and Sec. B in the appendix.

## 5 CONCLUSION

We introduced a computationally efficient, theoretically grounded uncertainty measure, the negative log-likelihood of the most likely output sequence under a given language model. This measure is motivated by the general notion of proper scoring rules, providing a theoretically justified measure that is well aligned with the practical usage of LLMs. The experiments show that our measure performs extremely well with just a single generated output sequence, compared to previous measures that require multiple costly sequences to estimate the uncertainty. As a result, our approach represents a significant advance toward providing reliable uncertainty estimates that can be effectively applied at scale.

Although our proposed measure effectively captures uncertainty, it currently does not consider the semantics of the generated output sequence. Future work should investigate how it could be extended to also account for semantic meaning, to further enrich the uncertainty estimator while preserving its computational efficiency.. Furthermore, all measures based on proper scoring rules depend on heuristics such as length normalization to deal with varying sequence lengths (Malinin & Gales, 2021; Duan et al., 2023; Bakman et al., 2024). Investigating theoretically justified means to account for these varying generation characteristics is another promising direction for future work. While there remain opportunities for refinement, our proposed measure establishes a solid foundation for reliable and scalable uncertainty estimation in NLG.

## ETHICS AND REPRODUCIBILITY STATEMENT

We acknowledge that language models can generate biased or harmful content if not properly managed. While our uncertainty estimation method enhances reliability, we encourage the responsible use of our approach in conjunction with bias mitigation and content moderation techniques.

We have made concerted efforts to ensure the reproducibility of our results. We report the raw average scores across held-out test datasets without standard error, as the distributional characteristics are more reflective of the models and datasets selected than the uncertainty estimation method itself. Theoretical derivations are provided in Sec. 2. All datasets are publicly available, and standard benchmarks are utilized to facilitate replication. The source code for reproducing all experiments will be made available upon publication.

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

## A    COMPARISION OF ESTIMATORS

In this section we want to empirically investigate the performance of estimators for the predictive entropy $H(p(\boldsymbol{y} \mid \boldsymbol{x}))$ (Eq. (5)) and the maximum likelihood $1 - \max_{\boldsymbol{y}} p(\boldsymbol{y} \mid \boldsymbol{x})$ (Eq. (8)). Therefore, we consider a synthetic experiment with the following setup. We are given a space of possible outcomes $\mathcal{V}$ with $|\mathcal{V}| = \{10, 100\}$. The task is to predict a sequence $\boldsymbol{y} = (y_1, ...y_T) \in \mathcal{Y}_T$ where $y \in \mathcal{V}$ and $T$ is 2, 3, or 4. Predictive distributions $p(\boldsymbol{y} \mid \boldsymbol{x}) = \prod_{t=1}^{T} p(\boldsymbol{y}_t \mid \boldsymbol{y}_{<t}, \boldsymbol{x})$ are not represented by a neural network, but randomly sampled (but fixed per run) according to a Dirichlet distribution $\text{Dir}(\{\alpha_1, ..., \alpha_{|\mathcal{V}|}\})$. The alpha parameters of the Dirichlet distribution are specified to yield typical predictive distributions as encountered in language models that follow a power law. For $|\mathcal{V}| = 10$ we have $\alpha_{1,2} = 10$ and $\alpha_{3-10} = 0.2$. For $|\mathcal{V}| = 100$ we have $\alpha_{1,2} = 10$, $\alpha_{3-6} = 1$ and $\alpha_{7-100} = 0.2$. Note that the order of alpha values is randomly shuffled before drawing each predictive distribution. Representative predictive distributions sampled from this Dirichlet distribution are shown in the leftmost subfigures in Fig. 2 and Fig. 3.

The experiments investigate the quality of the estimators depending on the number of samples $\{\boldsymbol{y}_n\}_{n=1}^{N}$. This is possible because it is possible to calculate the ground truth values for both the entropy and the maximum likelihood sequence for this small synthetic example by exhaustive enumeration. We average over 1,000 runs, meaning that the predictive distributions are redrawn according to the respective Dirichlet distribution. This corresponds to evaluating uncertainty for different input sequences $\boldsymbol{x}$ for language models.

**Entropy estimation.** The results are shown in Fig. 2. We observe that the variance of estimators increases for larger vocabulary sizes $|\mathcal{V}|$ and sequence lengths $T$. Furthermore, lower temperatures decrease the variance of the estimator at the cost of introducing bias.

**Maximum Likelihood.** The results are shown in Fig. 3. We observe that low-temperature multinomial sampling and beam search find the maximum log-likelihood with a low number of samples with high probability. Greedy decoding (beam size = 1) finds the maximum for all settings except the hardest ($\|\mathcal{V}| = 100, T = 4$), where it takes a beam size of 2 to find it.

## B    DETAILED RESULTS

In this section, we provide detailed results to complement the main results presented in Tab. 1.

The main results used greedy decoding (beam search of size 1) to estimate the maximum likelihood (zero-one score based measure) and 10 samples to estimate entropies (logarithmic score based measures). For each dataset, we performed a hyperparameter search on held-out instances to determine the best performing temperature $t \in \{0.5, 1.0, 1.5\}$ for sampling output sequences used for the logarithmic score based measures.

We look into how much the maximum likelihood benefits from additional samples by increasing the beam with to 5. The results are given in Tab. 2, showing that our measure continues to improve for a larger number of beams, thus better estimates of the maximum likelihood sequence. Furthermore, we provide detailed results for individual datasets in Tab. 3, complimenting the results presented in the main paper (c.f. Tab. 1).

The AUROC is considered as a primary performance measure throughout the paper. We additionally consider the average rejection accuracy, i.e. the accuracy of model predictions when allowing to reject a certain budget of predictions based on the uncertainty estimate. Thus, predictions are only evaluated for the 80% most certain predictions. Results are given in Tab. 4, again with greedy decoding for our measure based on the zero-one score. The results show, that our measure is very competitive across all settings, despite its simplicity and efficiency.

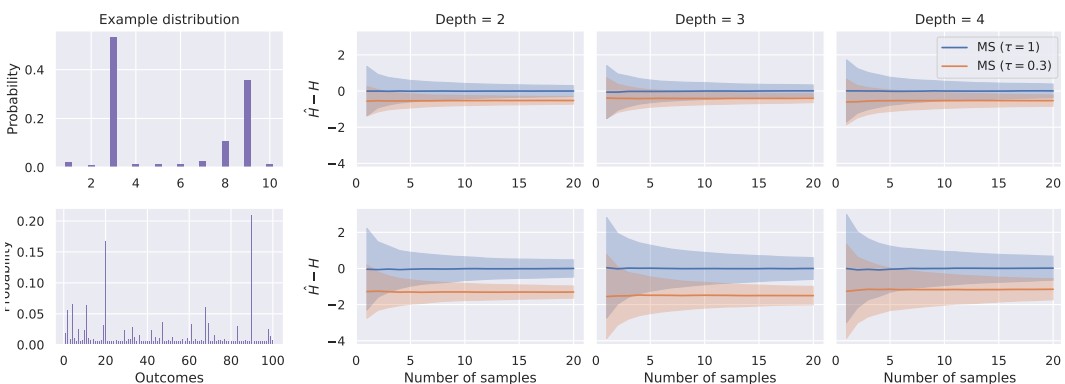

Figure 2: **Estimator of Predictive Entropy.** Results for different vocabulary sizes (rows) and sequence lengths (columns). The two leftmost subfigures show exemplary predictive distributions $p(\boldsymbol{y}_t \mid \boldsymbol{y}_{<t}, \boldsymbol{x})$. We estimate the entropy using $N$ samples by means of Eq. (5). Lines denote the average over runs, while shades denote one standard deviation. We compare multinomial sampling (MS) for two commonly used temperatures. The experiments show that temperature decreases variance but introduces bias.

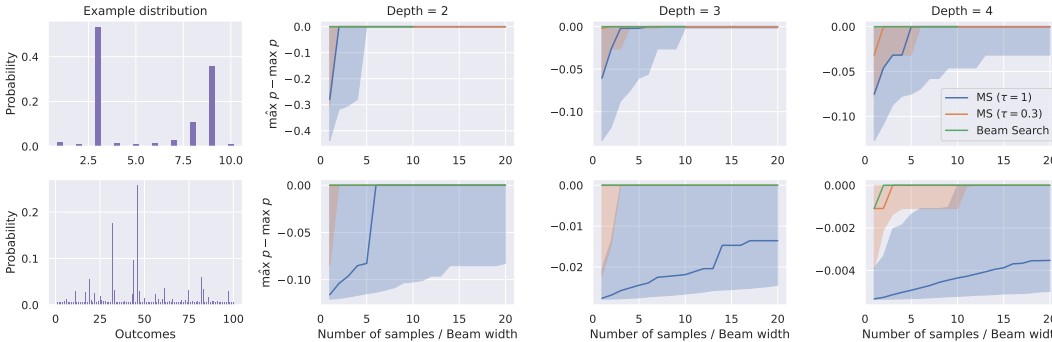

Figure 3: **Estimator of maximum likelihood.** Results for different vocabulary sizes (rows) and sequence lengths (columns). The two leftmost subfigures show exemplary predictive distributions $p(\boldsymbol{y}_t \mid \boldsymbol{y}_{<t}, \boldsymbol{x})$. We estimate the maximum likelihood using the maximum over $N$ sampled obtained by beam search or multinomial sampling (MS) with different temperatures. Lines denote the median, shades signify the possible values between the 5 and 95 percent quantile. Even with a very low number of samples, low-temperature multinomial sampling (MS) and beam search are able to find the maximum with high probability.

Table 2: **Average AUROC** across TriviaQA, SVAMP, and NQ datasets, using uncertainty estimates of different measures as a score to distinguish between correct and incorrect answers. Varying model architectures (*transformer*, *state-space*), model sizes (*7B*, *8B*, *70B*), and model stages (*PT*, *IT*) are considered for generating answers. The reference answer is generated using **beam search with 5 beams**, either as a whole sentence (*long*) or a short phrase (*short*). The correctness of the reference answer is assessed by checking if the F1 score of the commonly used SQuAD metric exceeds 0.5 (*F1*) or the Llama-3.1-70B model considers it as correct (*LLM*). Predictive Entropy (*PE*), length-normalized Predictive Entropy (*LN-PE*), Semantic Entropy (*SE*), length-normalized Semantic Entropy (*LN-SE*), and discrete Semantic Entropy (*D-SE*) use 10 output sequences to assign an uncertainty estimate, each generated via multinomial sampling. NLL solely uses the reference answer to assign an uncertainty estimate.

| *Uncertainty measure based score* | | | | *Logarithmic* | | | | | *Zero-One* |
|---|---|---|---|---|---|---|---|---|---|
| **Model** | | **Gen.** | **Metric** | **PE** | **LN-PE** | **SE** | **LN-SE** | **D-SE** | **NLL** |
| **Transformer** | **8B** PT | short | F1 | .775 | .791 | .765 | .787 | .799 | **.822** |
| | | short | LLM | .700 | .712 | .686 | .704 | .713 | **.726** |
| | | long | LLM | .556 | .540 | .493 | .520 | .578 | **.591** |
| | **8B** IT | short | F1 | .778 | .808 | .805 | .819 | .811 | **.845** |
| | | short | LLM | .682 | .704 | .706 | .713 | .698 | **.729** |
| | | long | LLM | .535 | .520 | .584 | .585 | **.586** | .559 |
| | **70B** PT | short | F1 | .788 | .799 | .796 | .812 | .798 | **.833** |
| | | short | LLM | .700 | .717 | .719 | **.727** | .718 | .725 |
| | | long | LLM | .540 | .552 | .489 | .531 | .552 | **.608** |
| | **70B** IT | short | F1 | .756 | .786 | .796 | **.806** | .788 | .800 |
| | | short | LLM | .680 | .697 | .701 | **.707** | .695 | **.707** |
| | | long | LLM | .534 | .533 | .544 | .569 | **.574** | .534 |
| **State-Space** | **7b** PT | short | F1 | .814 | .818 | .806 | .823 | .825 | **.846** |
| | | short | LLM | .703 | .709 | .699 | .711 | .712 | **.719** |
| | | long | LLM | .570 | .595 | .550 | **.609** | .602 | .563 |
| | **7b** IT | short | F1 | .799 | .815 | .794 | .817 | .828 | **.845** |
| | | short | LLM | .699 | .713 | .694 | .709 | .720 | **.730** |
| | | long | LLM | .574 | .575 | .582 | **.621** | .607 | .577 |

Table 3: **Average AUROC** of individual datasets, using uncertainty estimates of different measures as a score to distinguish between correct and incorrect answers.

| | | | | *Logarithmic* | | | | | *Zero-One* |
|---|---|---|---|---|---|---|---|---|---|
| *Uncertainty measure based score* | | | | | | | | | |
| $\mathcal{D}$ | **Model** | **Gen.** | **Metric** | **PE** | **LN-PE** | **SE** | **LN-SE** | **D-SE** | **NLL** |
| TriviaQA / Transformer / 8B | PT | short | F1 | .758 | .778 | .788 | .798 | .787 | **.810** |
| | PT | short | LLM | .675 | .694 | .703 | .704 | .682 | **.722** |
| | PT | long | LLM | .592 | .604 | .640 | .631 | .650 | **.704** |
| | IT | short | F1 | .735 | .768 | .790 | .800 | .777 | **.809** |
| | IT | short | LLM | .660 | .684 | .708 | .710 | .680 | **.716** |
| | IT | long | LLM | .603 | .627 | **.678** | .672 | .670 | .670 |
| TriviaQA / Transformer / 70B | PT | short | F1 | .707 | .730 | .741 | .743 | .702 | **.744** |
| | PT | short | LLM | .650 | .660 | .696 | .695 | .656 | **.698** |
| | PT | long | LLM | .538 | .533 | .625 | .574 | .563 | **.692** |
| | IT | short | F1 | .698 | .714 | .722 | **.726** | .688 | .722 |
| | IT | short | LLM | .663 | .675 | .685 | .679 | .633 | **.701** |
| | IT | long | LLM | .530 | .553 | .564 | **.571** | .564 | .543 |
| TriviaQA / State-Space / 7B | PT | short | F1 | .786 | .793 | .812 | .818 | .810 | **.832** |
| | PT | short | LLM | .687 | .697 | .712 | .714 | .695 | **.724** |
| | PT | long | LLM | .597 | .653 | .675 | .680 | .689 | **.705** |
| | PT | short | F1 | .780 | .799 | .810 | .819 | .811 | **.827** |
| | PT | short | LLM | .696 | .701 | .714 | .717 | .703 | **.730** |
| | PT | long | LLM | .645 | .654 | .688 | **.698** | .692 | .694 |
| SVAMP / Transformer / 8B | PT | short | F1 | .847 | .867 | .865 | .870 | .868 | **.885** |
| | PT | short | LLM | .779 | .788 | .753 | .772 | **.791** | .772 |
| | PT | long | LLM | .575 | .563 | .519 | .534 | .601 | **.669** |
| | IT | short | F1 | .879 | .903 | .914 | .912 | .887 | **.931** |
| | IT | short | LLM | .706 | .725 | .736 | .731 | .701 | **.753** |
| | IT | long | LLM | .556 | .524 | .590 | .608 | .631 | **.662** |
| SVAMP / Transformer / 70B | PT | short | F1 | .892 | .906 | .925 | .929 | .923 | **.936** |
| | PT | short | LLM | .794 | .817 | .814 | .815 | **.819** | .799 |
| | PT | long | LLM | .578 | .554 | .553 | .579 | .571 | **.665** |
| | IT | short | F1 | .830 | .895 | .915 | **.922** | .915 | .909 |
| | IT | short | LLM | .703 | .744 | .734 | .748 | **.762** | .713 |
| | IT | long | LLM | .601 | .577 | .613 | .649 | **.663** | .597 |
| SVAMP / State-Space / 7B | PT | short | F1 | .882 | .893 | .874 | .883 | .889 | **.914** |
| | PT | short | LLM | .752 | .757 | .730 | .738 | .757 | **.776** |
| | PT | long | LLM | .536 | .585 | .534 | .602 | **.612** | .579 |
| | IT | short | F1 | .843 | .891 | .854 | .876 | .892 | **.905** |
| | IT | short | LLM | .706 | .730 | .704 | .709 | .737 | **.744** |
| | IT | long | LLM | .577 | .586 | .578 | .616 | **.639** | .613 |
| NQ / Transformer / 8B | PT | short | F1 | .725 | .739 | .673 | .710 | .758 | **.776** |
| | PT | short | LLM | .639 | .661 | .615 | .641 | **.683** | **.683** |
| | PT | long | LLM | .517 | .498 | .478 | .495 | .550 | **.573** |
| | IT | short | F1 | .702 | .732 | .711 | .731 | .756 | **.774** |
| | IT | short | LLM | .662 | .682 | .669 | .685 | **.700** | .697 |
| | IT | long | LLM | .494 | .491 | **.530** | .524 | .527 | .514 |
| NQ / Transformer / 70B | PT | short | F1 | .727 | .733 | .711 | .737 | .748 | **.779** |
| | PT | short | LLM | .634 | .649 | .642 | .657 | .671 | **.672** |
| | PT | long | LLM | .538 | .514 | .494 | .553 | .580 | **.589** |
| | IT | short | F1 | .718 | .734 | .734 | **.748** | .746 | .743 |
| | IT | short | LLM | .676 | .674 | .689 | .698 | **.702** | .681 |
| | IT | long | LLM | .535 | .540 | .526 | .566 | **.574** | .545 |
| NQ / State-Space / 7B | PT | short | F1 | .766 | .758 | .741 | .765 | **.785** | .782 |
| | PT | short | LLM | .675 | .680 | .661 | .681 | **.697** | .683 |
| | PT | long | LLM | .567 | .553 | .512 | .551 | **.572** | .554 |
| | IT | short | F1 | .755 | .751 | .727 | .754 | **.783** | .781 |
| | IT | short | LLM | .669 | .672 | .648 | .671 | **.692** | .683 |
| | IT | long | LLM | .541 | .521 | .526 | .541 | **.554** | .537 |

Table 4: **Average Rejection Accuracy (80%)** across TriviaQA, SVAMP and NQ datasets, using uncertainty estimates of different measures as a score to distinguish between correct and incorrect answers. The reference answer is generated using greedy decoding, with the correctness being assessed by checking if the F1 score of the commonly used SQuAD metric exceeds 0.5 (*F1*), the pre-trained Llama-3.1-70B model considers it as correct (*LLM*), or the instruction-tuned Llama-3.1-70B-Instruct model considers it as correct (*LLM-Instruct*).

| *Uncertainty measure based score* | | | | *Logarithmic* | | | | | *Zero-One* |
|---|---|---|---|---|---|---|---|---|---|
| **Model** | | **Gen.** | **Metric** | **PE** | **LN-PE** | **SE** | **LN-SE** | **D-SE** | **NLL** |
| **Transformer** | **8b** | | | | | | | | |
| | | short | F1 | .661 | .672 | .651 | .643 | .655 | **.681** |
| | | short | LLM | .774 | **.782** | .767 | .766 | .765 | .778 |
| | PT | | LLM-Instruct | .704 | .721 | .693 | .688 | .702 | **.723** |
| | | long | LLM | .596 | .590 | .598 | .592 | .590 | **.619** |
| | | long | LLM-Instruct | .667 | .684 | .632 | .643 | .644 | **.686** |
| | | short | F1 | .668 | .684 | .680 | .673 | .687 | **.702** |
| | | short | LLM | .775 | .781 | .779 | .775 | .778 | **.788** |
| | IT | | LLM-Instruct | .723 | .742 | .732 | .726 | .743 | **.751** |
| | | long | LLM | .628 | .630 | .651 | .644 | **.653** | .652 |
| | | long | LLM-Instruct | .713 | .724 | .705 | .713 | .727 | **.734** |
| | **70b** | short | F1 | .818 | .827 | .822 | .827 | .829 | **.836** |
| | | short | LLM | .844 | .852 | .846 | .847 | .851 | **.855** |
| | PT | | LLM-Instruct | .867 | .875 | .876 | .881 | **.885** | .881 |
| | | long | LLM | .704 | .699 | .719 | .707 | .705 | **.724** |
| | | long | LLM-Instruct | .789 | .795 | .776 | .781 | .788 | **.812** |
| | | short | F1 | .795 | .813 | .814 | .809 | .819 | **.823** |
| | | short | LLM | .836 | .842 | .842 | .837 | .844 | **.845** |
| | IT | | LLM-Instruct | .850 | .867 | .866 | .865 | **.874** | .870 |
| | | long | LLM | .706 | .706 | .712 | .715 | **.721** | .715 |
| | | long | LLM-Instruct | .855 | .850 | .827 | .842 | **.861** | .851 |
| **State-Space** | **7b** | short | F1 | .598 | .596 | .585 | .579 | .583 | **.612** |
| | | short | LLM | .729 | .737 | .723 | .721 | .733 | **.742** |
| | PT | | LLM-Instruct | .638 | .640 | .626 | .621 | .632 | **.651** |
| | | long | LLM | .613 | **.627** | .612 | .624 | .620 | .623 |
| | | long | LLM-Instruct | .606 | .611 | .601 | .611 | .618 | **.633** |
| | | short | F1 | .592 | .603 | .588 | .581 | .589 | **.615** |
| | | short | LLM | .737 | .742 | .730 | .726 | .740 | **.744** |
| | IT | | LLM-Instruct | .632 | .646 | .625 | .619 | .637 | **.653** |
| | | long | LLM | .611 | .617 | .618 | .612 | **.625** | .625 |
| | | long | LLM-Instruct | .643 | .652 | .628 | .628 | .654 | **.658** |