# OpenReview forum: "Rethinking Uncertainty Estimation in Natural Language Generation"
_ICLR.cc/2025/Conference — Submitted to ICLR 2025_

### Official Review · Reviewer_jdx2 · 2024-11-02

**Soundness:** 2
**Presentation:** 1
**Contribution:** 1
**Rating:** 3
**Confidence:** 4

**Summary:**

The paper studies the uncertainty quantification for LLM. The paper first defines uncertainty as the expected score (which needs to be designed) of LLM prediction with respect to all possible parameters fitting the data. It then defines the scoring function as zero-one indicator of whether the generated sequence reaches maximal likelihood. The paper claims the estimation of the final uncertainty quantity only requires max-decoding (such as beam search). The paper evaluates the proposed method against prior baselines on 3 tasks for 6 LLMs. The paper used AUROC to measure accuracy and claims the proposed method achieves the best.

**Strengths:**

1. The uncertainty quantification for LLM is an important topic.
2. The derivation of aleatoric and epistemic entropy is valid.
3. The introduction of zero-one score is valid.

**Weaknesses:**

1. The writing of the paper is very blurred. It is not clear which part is from prior paper, which part is the original contribution in this paper. Eq(7-8) are proposed in this paper. But Eq(1-6) are unclear.
2. The definition of uncertainty in Eq (1) seems to suggest there is a groundtruth y generation. The definition in Eq(2) is questionable. It is unclear what is the posterior distribution of parameter w. Does it need to have a distribution of parameter? What if the parameter is fixed.
3. The actual estimation algorithm is not described. In particular, Eq(8) needs to estimate an expectation term. It is unclear how to estimate this part. There is no description in the paper.
4. The evaluation approach and the metric used are quite questionable. It is unclear why this particular F1 threshold-based correctness is used. But the details of estimation such correctness is also not described well. The use of LLaMA 70B as the evaluator is also quite questionable.

**Questions:**

1. Are eq (1-6) developed by you or prior work?
2. what is the exact step to calculate Eq(8).
3. Do you have real groundtruth measurement for generation correctness?
4. How many generations do you need to estimate uncertainty for one sequence?

---

> ### Author Response · Authors · 2024-11-25
>
> Thank you for your rebuttal and for acknowledging the validity of our theoretical results. We have addressed your concerns and refined the manuscript further, as detailed below:
>
> ---
> > The writing of the paper is very blurred. It is not clear which part is from prior paper, which part is the original contribution in this paper. Eq(7-8) are proposed in this paper. But Eq(1-6) are unclear. *Related 1. Question*: Are eq (1-6) developed by you or prior work?
>
> We acknowledge any lack of clarity in the original submission and have made substantial revisions to enhance the paper’s readability and overall presentation in the updated version.
>
> To clarify, you are correct that Eq. (7-8) are proposed in this paper. Eq. (1-2) introduce the proper scoring rules, which are established concepts from prior work. Similarly, Eq. (3-6) are also derived from prior work, as indicated by their placement in the subsection titled “Established Uncertainty Measures”.
>
> ---
> > The definition of uncertainty in Eq (1) seems to suggest there is a groundtruth y generation. The definition in Eq(2) is questionable. It is unclear what is the posterior distribution of parameter w. Does it need to have a distribution of parameter? What if the parameter is fixed.
>
> Eq. (1) represents the general framework of proper scoring rules and does not assume the existence of a “ground-truth generation.” Instead, $y’$ refers to any observed output sequence, while $p(y | x, \cdot)$ is the probability distribution over possible output sequences.
>
> Regarding Eq. (2), the posterior distribution $p(w | \mathcal{D})$  is well-defined in a Bayesian context, representing our belief over possible parameter values given the training data. This posterior enables the quantification of epistemic uncertainty by assessing the divergence between predictions under different plausible parameter values.
>
> ---
> > The actual estimation algorithm is not described. In particular, Eq(8) needs to estimate an expectation term. It is unclear how to estimate this part. There is no description in the paper. *Related 2. Question*: what is the exact step to calculate Eq(8).
>
> There seems to be a misunderstanding regarding Eq. (8). As indicated in the paper, our focus is on the aleatoric uncertainty, which solely considers the most likely output sequence. The expectation term pertains to the epistemic uncertainty. We approximate the most likely output sequence using greedy decoding, a practical and computationally efficient approach.
>
> ---
> > The evaluation approach and the metric used are quite questionable. It is unclear why this particular F1 threshold-based correctness is used. But the details of estimation such correctness is also not described well. The use of LLaMA 70B as the evaluator is also quite questionable.
>
> The F1 threshold-based correctness metrics are indeed common practice in the field of uncertainty estimation and align with widely accepted benchmarks.
> To complement the statistics-based F1 metric, we selected LLaMA 70B for the LLM-as-a-Judge metric due to its superior capabilities among the models we evaluated.
>
> While we acknowledge that automated evaluation has limitations, our approach aligns with the evaluation of recent uncertainty estimation literature and provides a practical and consistent framework for assessing the correctness of answers across different tasks.
>
> ---
> > How many generations do you need to estimate uncertainty for one sequence?
>
> Our method estimates uncertainty using only a single output sequence (generated via greedy decoding). In contrast, previous methods, such as Predictive Entropy or Semantic Entropy, typically require generating multiple output sequences (commonly around 10), making our approach significantly more computationally efficient while maintaining competitive performance.
>
> ---
> Overall, we hope that our clarifications address your concerns and strengthen your view of our work. Should you have any further questions, we look forward to addressing them. Otherwise, we hope for a positive reassessment of our work.

---

> > ### Comment · Reviewer_jdx2 · 2024-11-25
> >
> > I read and appreciate the authors' response.
> >
> > 1. Regarding the prior, is it possible to define a prior for a 7B LLM's parameters (or even larger) in an effective and meaningful way?
> >
> > 2. Using max-decoding to approximate the expectation requires certain assumption. It is a big assumption.

---

> > > ### Author Response · Authors · 2024-11-26
> > >
> > > Thank you for the reply!
> > >
> > > ---
> > > 1. Specifying meaningful priors for large language models is indeed a challenging task, as highlighted in the recent position paper on Bayesian Deep Learning by Papamarkou et al. [1] (Section 3.4), which provides an overview of recent advances in prior specification. However, it is important to note that our work's focus is not on posterior inference but on estimating aleatoric uncertainty, as we work with a single LLM. We have further emphasized this focus in the revised version of our paper.
> > >
> > > ---
> > > 2. Our proposed uncertainty measure is the aleatoric component of Eq. (8), which does not involve an expectation but instead is the log-likelihood of the greedily decoded output sequence. To clarify this further, we have added Eq. (9) and expanded the discussion in the revised version of our paper to ensure this point is explicit.
> > >
> > > ---
> > > We hope that the substantial updates to our paper resolve your concerns and lead to a favorable reassessment of our work.
> > >
> > > ---
> > > [1] Papamarkou et al. (2024) Position: Bayesian Deep Learning is Needed in the Age of Large-Scale AI, ICML

---

> ### Comment · Reviewer_jdx2 · 2024-12-03
>
> I appreciate authors' effort in revising the paper. However, Section 2 is still mixing background, prior work and new work. It is still not clear which part is the new contribution (It seems Sec 2.2 is newly proposed one). I would encourage the authors split this section into two. One for the problem setup and background. The other is for the proposed method.
>
> The authors may add justification and an example of a proper prior for LLM's parameters.
>
> I am skeptical about LLaMA's capability in judging (without finetuning). The authors may add additional evaluation using a larger model (e.g. GPT4, Gemini, Claude), or add human evaluation to validate.
>
> The main method part also needs some leap of faith. The authors may add additional explanations of assumptions and justification of them.

---

### Official Review · Reviewer_obDG · 2024-11-03

**Soundness:** 1
**Presentation:** 1
**Contribution:** 2
**Rating:** 3
**Confidence:** 3

**Summary:**

This paper proposes to quantify the uncertainty of a language model for a specific prompt using the log-likelihood of the most probable sequence. Empirical results show that this new measure is effective in quantifying the uncertainty of the model without having to generate multiple times.

**Strengths:**

1. The paper studies an important topic, which is crucial for many applications (e.g. improving trustworthiness of LMs).

2. The experiment results are good, which is surprising given the simplicity of the proposed method.

**Weaknesses:**

1. The idea to use a single generation to “approximate the most likely output sequence (line 223)” is concerning - the motivation is to avoid generating multiple sentences, and yet in order for beam search to find the most probable sentence (even in a toy setting), it requires multiple samples (Appendix A, Figure 2). Practically, I don’t know how close a greedy sampled / top-k sampled sequence is close to the most likely sequence even of the same length.

2. The contribution (using log-likelihood of one generation) is somewhat limited to empirical findings without any theoretical guarantees that one generation is able to find a sequence that is close to the most probable sequence.

2. The paper is not very well written, making it difficult to understand what the authors want to convey. See the questions section.

**Questions:**

083: What is \mathcal{V} here? You need to introduce the vocabulary.

094: “since \mathcal{Y} scales exponentially with the sequence length T.” Here you defined \mathcal{Y} as all possible sequences, which is an infinite set so it shouldn’t be growing. If you want to make this claim, you can define \mathcal{Y}_t as the subset of all sequences with length <t.

098-100: “We consider uncertainty for a given LMs, … a valid assumption” I am a bit confused, what is the assumption here?

111: Here if you are sampling y’ from p(\cdot |x, \cdot) it is better to write it explicitly “y’ \sim …” It can be a bit confusing here, and I am not sure how the discussion on “Proper Scoring Rules for Uncertainty Measures.” advances your main claims - if this is only about evaluation, you can defer this to later sections.

127: Again, I am not sure how “Aleatoric and Epistemic Uncertainty” relates to your proposed method. The purpose of “related works” or “preliminaries” is to make people ground your work to existing literature, if you believe that your proposed method is connected with this literature, make it more explicit.

898: “The reference answer sampled using beam search with a size of 20 is considered for assessing overall correctness, as it represents the most likely answer generated by the language model” - why is beam search of 20 = most probable answer? Do you have any guarantees that a beam size of x makes the generated sequence log-prob close (difference bounded) to the most likely sequence?

---

> ### Author Response · Authors · 2024-11-25
>
> Thank you for the valuable feedback and for acknowledging the importance and effectiveness of our method. We have addressed your concerns and refined our paper, as detailed below:
>
> ---
> > 1. The idea to use a single generation to “approximate the most likely output sequence (line 223)” is concerning - the motivation is to avoid generating multiple sentences, and yet in order for beam search to find the most probable sentence (even in a toy setting), it requires multiple samples (Appendix A, Figure 2). Practically, I don’t know how close a greedy sampled / top-k sampled sequence is close to the most likely sequence even of the same length.
>
> Thank you for raising this important point. In our experiments, we used greedy decoding as a practical approximation to the most likely output sequence. As shown in Figure 3 in the appendix, greedy decoding achieves performance comparable to more precise methods, such as beam search with up to 20 beams, empirically supporting its validity as a proxy for the most likely output sequence. In contrast, multinomial sampling significantly degrades the performance of the uncertainty measure, further emphasizing the robustness of greedy decoding in this context. Importantly, greedy decoding incurs the same computational cost as sampling a single sequence. We have clarified this in the revised version of the paper.
>
> ---
> > 2. The contribution (using log-likelihood of one generation) is somewhat limited to empirical findings without any theoretical guarantees that one generation is able to find a sequence that is close to the most probable sequence.
>
> We acknowledge that finding the most likely output sequence is theoretically challenging due to the exponential growth of the output space. However, greedy decoding effectively approximates the most likely output sequence under conditions of low-entropy token distributions, where divergences can be probabilistically bounded. Our empirical results, as outlined above, strongly suggest that current language models meet these conditions, supporting the practicality of this approach.
>
> ---
> > 3. The paper is not very well written, making it difficult to understand what the authors want to convey. See the questions section.
>
> We regret any lack of clarity in the original submission and have made significant efforts to improve the paper’s readability and presentation in the revised version.
>
> ---
> **Questions**
>
> Thank you for pointing out all the questions and suggestions. We revised the paper accordingly, by explicitly introducing the vocabulary, rigorously defining the set of all output sequences, and further elaborating on the uncertainty measure used.
>
> ---
> > I am not sure how the discussion on “Proper Scoring Rules for Uncertainty Measures.” advances your main claims - if this is only about evaluation, you can defer this to later sections.
>
> The discussion on “Proper Scoring Rules for Uncertainty Measures” is fundamental for deriving the negative log-likelihood of the most likely output sequence as a proper uncertainty measure. Since it seems that this message has not come across clear enough, we made this important contribution explicit in the revised version of the paper.
>
> ---
> >  Why is beam search of 20 = most probable answer? Do you have any guarantees that a beam size of x makes the generated sequence log-prob close (difference bounded) to the most likely sequence?
>
> Among the decoding strategies we evaluated (multinomial sampling, greedy decoding, and beam search), the output from beam search with the highest number of beams (20) provides the best practical approximation to the most likely output sequence. Thus, we use this sequence as the reference for assessing uncertainty. This approach aligns with common practice in related work, where the best approximation to the most likely sequence is typically used as the reference.
>
> ---
> Overall, we hope that our clarifications and revisions address your concerns and further strengthen your view of our work. Should you have any further questions, we look forward to addressing them. Otherwise, we hope for a positive reassessment of our work.

---

### Official Review · Reviewer_QzXV · 2024-11-04

**Soundness:** 2
**Presentation:** 2
**Contribution:** 2
**Rating:** 3
**Confidence:** 3

**Summary:**

Traditional uncertainty estimation relies on sampling-based methods, which inevitably incurs additional computation cost. This work addresses this limitation and proposes to measure the uncertainty solely based on the negative log-likelihood of the most likely sequence. Empirical results demonstrate the performance of the proposed metric in distinguishing between correct and incorrect answers.

**Strengths:**

1. The proposed metric alleviates the need for sampling to estimate the uncertainty in natural language generation.

2. The derivation of the different uncertainty terms and defining aleatoric and epistemic uncertainty is helpful.

3. The presented experiments cover a few backbone models and representative tasks.

**Weaknesses:**

1. The only metric proposed by this work is the zero-one score, which is one minus the predictive distribution for the most likely output sequence. Therefore, I find this is actually equivalent to propose $p(y=y’|x)$ as the confidence estimation, which has been widely applied in the machine learning community, whereas uncertainty is simply derived by 1-confidence. Consequently, this metric lacks technical novelty.

2. Though the proposed NLL metric seems to be superior to baselines, this work lacks justification and insights on why the NLL is a better metric than the variants using sampling.

3. Verbal explanations have been widely implemented in estimating the confidence level of LLMs. The author includes relevant discussion in Line 249. However, there is no empirical comparison to this type of baseline.

**Questions:**

1. What’s $\mathcal{D}$ in Line 115?

2. Could you provide more justifications on why NLL may be better than other sampling-based baselines?

---

> ### Author Response · Authors · 2024-11-25
>
> Thank you for your insightful comments and for acknowledging the theoretical contributions and effectiveness of our method. Below, we would like to address your concerns:
>
> ---
> > The only metric proposed by this work is the zero-one score, which is one minus the predictive distribution for the most likely output sequence. Therefore, I find this is actually equivalent to propose as the confidence estimation, which has been widely applied in the machine learning community, whereas uncertainty is simply derived by 1-confidence. Consequently, this metric lacks technical novelty.
>
> You are correct that our metric aligns with the concept of *confidence estimation*. This is precisely the main contribution of our paper: demonstrating that, **under the right assumptions**, *confidence estimation* serves as a proper and theoretically justified uncertainty measure. To our knowledge, this connection has not been explicitly established before. While *confidence estimation* is widely applied in the broader machine learning community, this simple yet effective approach is often overlooked and rarely used as a baseline for uncertainty estimation in NLG. By rigorously deriving its theoretical foundations and empirically validating its performance, we challenge the complexity of current uncertainty estimation methods and offer a computationally efficient alternative.
>
> ---
> > Though the proposed NLL metric seems to be superior to baselines, this work lacks justification and insights on why the NLL is a better metric than the variants using sampling. *Related Question:* Could you provide more justifications on why NLL may be better than other sampling-based baselines?
>
> While uncertainty measures based on the logarithmic score would presumingly be advantageous if the full distribution over output sequences  $p(y \mid x, w)$ was accessible – similar to classification tasks – this distribution is intractable for NLG tasks. As a result, sampling-based methods can only provide crude approximations, often limited by computational constraints and sampling variability. In contrast, uncertainty measures based on the zero-one score, such as the NLL metric, offer a more rigorous and computationally efficient alternative by leveraging the model’s inherently well-calibrated confidence, without relying on extensive sampling.
>
> ---
> > Verbal explanations have been widely implemented in estimating the confidence level of LLMs. The author includes relevant discussion in Line 249. However, there is no empirical comparison to this type of baseline.
>
> Thank you for pointing this out. While verbal explanations are a valuable heuristic for assessing confidence levels in LLMs, they lack a theoretical foundation in aleatoric uncertainty estimation. For this reason, we focused our comparisons on uncertainty measures that are derived from well-established theoretical principles, within the constraints of our computational budget (>3000 A100 GPU hours).
>
> It is important to note that we do not claim that the NLL metric is universally superior. Instead, we demonstrate that this simple and computationally efficient metric performs better than current state-of-the-art uncertainty measures grounded in the same principles. By emphasizing its theoretical rigor and strong empirical performance, we aim to highlight the NLL metric as a practical alternative to sampling-based uncertainty measures in NLG.
>
> ---
> > **Question:** What’s $\mathcal{D}$ in Line 115?
>
> Thank you for pointing this out. $\mathcal{D}$ refers to the fixed training data. We have formally introduced this notation in the revised version of the paper to ensure clarity.
>
> ---
> Overall, we hope that our clarifications address your concerns and strengthen your view of our work. Should you have any further questions, we look forward to addressing them. Otherwise, we hope for a positive reassessment of our work.

---

### Official Review · Reviewer_meYg · 2024-11-04

**Soundness:** 3
**Presentation:** 3
**Contribution:** 3
**Rating:** 5
**Confidence:** 3

**Summary:**

The paper presents a MAP-based approach to estimating uncertainty in large language models (LLMs) to improve the reliability of generated text. Traditional Monte-Carlo uncertainty estimation methods rely on generating multiple output sequences, a process that is computationally intensive and inefficient at scale. This study introduces a streamlined method that estimates uncertainty using only the negative log-likelihood of the most probable output sequence, eliminating the need for multiple sequences. The proposed approach maintains theoretical rigor and outperforms or matches existing methods across a range of tasks and models.

**Strengths:**

- Computational Efficiency: The proposed uncertainty measure requires only a single output sequence, significantly reducing computational costs compared to methods that generate multiple sequences, making it highly scalable for real-world applications.
- Theoretical Soundness: Using MAP as the metric of uncertainty is grounded in established principles of proper scoring rules, ensuring theoretical robustness while simplifying the complexity of uncertainty estimation for natural language generation models​.

**Weaknesses:**

- Questionable Efficiency: It seems to me that obtaining the MAP sequence (argmax) is non-trivial. While seemingly at the end of the day we only get one sequence, taking efforts to approximate it to be the MAP could be no computationally cheaper than sampling a lot of candidates, which is exactly what the paper is claiming to avoid. It would make the paper more compelling, if the authors can briefly study how well the argmax sequence is approximated, and if the approximation of obtaining argmax is not quite good, what is the worst-case performance of the proposed method.

**Questions:**

Please refer to Weaknesses).

---

> ### Author Response · Authors · 2024-11-25
>
> We appreciate your constructive feedback, especially for highlighting the theoretical soundness of our work and the computational efficiency of our method. Below, we would like to address your raised concern:
>
> ---
> > Questionable Efficiency: It seems to me that obtaining the MAP sequence (argmax) is non-trivial. While seemingly at the end of the day we only get one sequence, taking efforts to approximate it to be the MAP could be no computationally cheaper than sampling a lot of candidates, which is exactly what the paper is claiming to avoid. It would make the paper more compelling, if the authors can briefly study how well the argmax sequence is approximated, and if the approximation of obtaining argmax is not quite good, what is the worst-case performance of the proposed method.
>
> This is indeed a very valuable point. In our experiments, we used greedy decoding as a practical approximation to the most likely output sequence. As shown in Figure 3 in the appendix, greedy decoding performs comparably to more precise approximations using up to 20 beams, supporting its validity as a proxy for the most likely output sequence. In contrast, multinomial sampling with temperatures of $0.5$ or $1.0$ significantly degrades the performance of the uncertainty measure. We have clarified this point further in the revised version of the paper. Importantly, greedy decoding incurs the same computational cost as sampling a single sequence, aligning with the computational efficiency highlighted as a strength of our approach.
>
> ---
> Overall, we hope that our clarifications and revisions address your concerns and further strengthen your view of our work. Should you have any further questions, we look forward to addressing them. Otherwise, we hope for a positive reassessment of our work.

---

### Official Review · Reviewer_mosu · 2024-11-04

**Soundness:** 2
**Presentation:** 3
**Contribution:** 3
**Rating:** 6
**Confidence:** 3

**Summary:**

This paper presents a novel approach to uncertainty estimation in natural language generation (NLG) models. The authors propose using the negative log-likelihood (NLL) of the generated sequence as a surrogate for uncertainty estimation. By leveraging the theoretical framework of proper scoring rules, they demonstrate that NLL can serve as an effective uncertainty metric. This approach simplifies the estimation process because it only requires the likelihood of the generated sequence under the model, avoiding the need for multiple samples. The theoretical foundation is well-established within the framework of proper scoring rules, and the empirical results demonstrate the method's superiority over existing metrics across various models and tasks.

**Strengths:**

- The paper introduces a new uncertainty estimation metric based on NLL that avoids the need for multiple sequence generations, which is a common bottleneck in existing methods.
- By eliminating the need to generate multiple output sequences, the proposed method significantly reduces computational overhead, making it more practical for large-scale applications.
- The method achieves or surpasses the performance of existing state-of-the-art uncertainty estimation methods across different models and tasks.
- The approach shows strong performance across various model architectures, sizes, and training stages, demonstrating its broad applicability.

**Weaknesses:**

- The proposed metric focuses on statistical uncertainty derived from model probabilities but does not explicitly account for the semantic aspects of generated text. Incorporating semantic uncertainty would provide a more holistic estimation, capturing discrepancies between the generated content and the underlying meaning or intent. While the authors briefly discuss this limitation in the conclusion, it remains a significant issue that warrants deeper exploration, possibly through additional methods or combined metrics.
- While the experiments are extensive, they focus primarily on free-form question-answering tasks. Additional experiments on other types of NLG tasks (e.g., dialogue generation, story generation) would strengthen the claims.
- The paper spans 7 pages, whereas the conference allows submissions up to 10 pages. This unused space represents an opportunity to expand on key areas such as additional experiments, detailed analyses, or discussions that could further strengthen the paper's contributions.

**Questions:**

N/A

---

> ### Author Response · Authors · 2024-11-25
>
> We thank you for the thoughtful feedback and for recognizing our method’s computational practicality for large-scale applications and its state-of-the-art performance across diverse models and tasks. Below, we address your comments and concerns:
>
> ---
> > The proposed metric focuses on statistical uncertainty derived from model probabilities but does not explicitly account for the semantic aspects of generated text. Incorporating semantic uncertainty would provide a more holistic estimation, capturing discrepancies between the generated content and the underlying meaning or intent. While the authors briefly discuss this limitation in the conclusion, it remains a significant issue that warrants deeper exploration, possibly through additional methods or combined metrics.
>
> While we acknowledge the importance of semantic uncertainty in many contexts, our work intentionally focuses on challenging the prevailing assumption that semantic clustering is essential for state-of-the-art uncertainty estimation in NLG. This shift in perspective is a central contribution of our paper, as reflected in its title, “Rethinking Uncertainty Estimation”.
>
> By demonstrating that uncertainty can be effectively estimated using a single output sequence, our approach paves the way for more efficient and practical methods. We recognize that future work could extend our method to also incorporate semantic aspects, further enriching uncertainty estimation while preserving its computational efficiency.
>
> ---
> > While the experiments are extensive, they focus primarily on free-form question-answering tasks. Additional experiments on other types of NLG tasks (e.g., dialogue generation, story generation) would strengthen the claims.
>
> We agree that additional experiments on other types of NLG tasks could further broaden the evaluation. However, we intentionally aligned our experiments with prior work that primarily focuses on question-answering tasks [1,2,3,4] to ensure a fair and consistent comparison of uncertainty estimation methods within the constraints of our computational budget (>3000 A100 GPU hours).
>
> It is worth noting that the recent Nature publication on semantic entropy [2] additionally evaluated longer paragraph-length generations by breaking them down into individual question-answering tasks. This suggests that performance on question-answering tasks correlates strongly with performance on more extensive NLG scenarios, reinforcing the validity of our chosen evaluation framework.
>
> ---
> > The paper spans 7 pages, whereas the conference allows submissions up to 10 pages. This unused space represents an opportunity to expand on key areas such as additional experiments, detailed analyses, or discussions that could further strengthen the paper's contributions.
>
> Thank you for this valuable feedback. Our goal was to maintain a concise presentation in the main text, focusing on the core insights, while including more detailed analyses in the appendix for thoroughness. Based on your suggestion, we have expanded the main text to incorporate additional discussion and provide greater context for our contributions.
>
> ---
> Overall, we hope that our clarifications address your concerns and further strengthen your view of our work. Should you have any further inquiries, we look forward to addressing them. Otherwise, we hope for a positive reassessment of our work.
>
> ---
> [1] Lorenz Kuhn, Yarin Gal, Sebastian Farquhar. Semantic Uncertainty: Linguistic Invariances for Uncertainty Estimation in Natural Language Generation. arXiv, 2302.09664, 2023
>
> [2] Sebastian Farquhar, Jannik Kossen, Lorenz Kuhn, and Yarin Gal. Detecting hallucinations in large language models using semantic entropy. Nature, 2024
>
> [3] Jinhao Duan, Hao Cheng, Shiqi Wang, Alex Zavalny, Chenan Wang, Renjing Xu, Bhavya Kailkhura, and Kaidi Xu. Shifting Attention to Relevance: Towards the Predictive Uncertainty Quantification of Free-Form Large Language Models. arXiv, 2307.01379, 2023
>
> [4] Alexander Nikitin, Jannik Kossen, Yarin Gal, Pekka Marttinen. Kernel Language Entropy: Fine-grained Uncertainty Quantification for LLMs from Semantic Similarities. arXiv, 2405.20003, 2024

---

> > ### Comment · Reviewer_mosu · 2024-12-03
> > **Response to authors**
> >
> > Thank you for the clarification. I have carefully reviewed your response as well as the comments from the other reviewers during the rebuttal phase. I have decided to maintain my original score.

---

### Author Response · Authors · 2024-11-26

Dear Reviewers,

We have worked diligently to substantially extend and improve the paper based on your valuable feedback.

As the end of the rebuttal phase is approaching, we kindly invite you to take a look at the revised version of our paper and our detailed responses. If you have any additional questions or require further clarification, we would be more than happy to address them.

Thank you once again for your insightful comments and the effort you have dedicated to reviewing our work.

---

> ### Author Response · Authors · 2024-12-01
>
> Dear Reviewers,
>
> With the rebuttal phase coming to a close, we kindly follow up to check whether our extensive paper revisions and detailed responses have addressed your questions and concerns. If so, we would be grateful for this to be reflected in your final assessment of our work. Should any points remain, we would be happy to discuss them during the final day of the rebuttal phase.
>
> Thank you!

---

### Meta-Review · Area_Chair_CdB5 · 2024-12-21

**Metareview:**

Based on the reviewers' feedback, I recommend rejecting this paper at this time. While the paper presents an interesting approach to uncertainty estimation in LLMs using negative log-likelihood of the most likely sequence, several fundamental issues emerge: theoretical concerns about obtaining the MAP sequence (which may be no cheaper than sampling multiple candidates), limited technical novelty as the proposed metric is essentially equivalent to commonly used confidence estimation, and unclear algorithmic details particularly around expectation term estimation. The writing is notably unclear about distinguishing prior work from novel contributions. Additionally, the evaluation methodology raises questions, especially regarding the F1 threshold-based correctness metric and the use of LLaMA 70B as evaluator. Without addressing these theoretical foundations and methodological concerns, the paper's contributions remain inadequately demonstrated.

**Additional Comments On Reviewer Discussion:**

I have read the messages in the discussion period and my opinion has been summarized as in the metareview above. I considered these points in my recommendation.

---

### Decision · Program_Chairs · 2025-01-22

Reject